# Investigation on Performances and Functions of Asphalt Mixtures Modified with Super Absorbent Polymer (SAP)

**DOI:** 10.3390/ma16031082

**Published:** 2023-01-26

**Authors:** Yuxuan Sun, Weimin Song, Hao Wu, Yiqun Zhan, Zhezheng Wu, Jian Yin

**Affiliations:** 1School of Civil Engineering, Central South University, Changsha 410017, China; 2School of Civil Engineering & Mechanics, Central South University of Forestry and Technology, Changsha 410004, China

**Keywords:** superabsorbent polymer, asphalt materials, performances, functions, water infiltration

## Abstract

The super absorbent polymer (SAP) has been attracting extensive concerns due to its strong capacity in water absorption and retention. The amorphous hydrogels formed by the post-absorbent SAP have the potential of clogging the micro-cracks in asphalt materials and refraining the rainwater from infiltrating. This provides the possibility of applying SAP in asphalt pavements to seal or fill the cracks and relieve the distresses caused by rainwater infiltration in the underlying layers. Before exploring the cracking sealing mechanism of SAPs in asphalt pavements, a series of experiments were performed to evaluate the feasibility and influences of SAPs in asphalt mastics and asphalt mixtures on their mechanical performances and functionalities. Firstly, the basic properties of SAPs were analyzed, and then the rheological properties of the asphalt mastics using SAP replacing mineral powder (10%, 20%, 30%, and 40% by volume) were explored. The water stability and infiltration reduction effect of the asphalt mixtures incorporated with SAP were evaluated by the Marshall stability test, immersion Marshall stability test, freeze-thaw splitting strength test, Cantabro test, and permeability test. The test results indicated that SAPs could be used in the asphalt mixtures to partially substitute mineral powder with desirable mechanical performances. When less than 10% of the mineral powder was replaced by the SAP, the high-temperature performance and fatigue life of the asphalt mastics could be improved to some extent, but both declined after the content of the SAP was larger than 10%. Due to the hydrogels formed by SAPs after water absorption, the water stability of the asphalt mixtures deteriorated with the increased content of SAPs. Moreover, the results from the permeability tests implied that the SAP hydrogels could fill the seepage channels in the material, thus improving the migration and infiltration resistances of the asphalt mixtures. With the increased contents of SAPs, the permeability coefficients of the asphalt mixtures could be reduced up to 55%. Based on the research findings in this study, when an appropriate amount of SAP was added in the asphalt materials, desirable temperature stability, water stability, and fatigue resistance could be achieved regarding actual requirements from applications. At the same time, the addition of SAPs could effectively refrain the infiltration and migration of rainwater in asphalt pavements, thus potentially mitigating the effect of water erosion on the underlying layers.

## 1. Introduction

Asphalt pavements became the first choice for highway construction because of their excellent mechanical properties, desirable smoothness and roughness, fast construction speed, and convenient maintenance [1,2,3]. However, asphalt pavements can be easily affected by the rainwater, pavement surface water, and other kinds of moisture in their service lives. Many studies [4,5,6,7] have reported that moisture would invade the interior of the pavement structure through the aggregate gap, pavement surface micro-cracks, construction joints, and other ways. The micro-cracks produced in the initial stage of the pavement have no obvious influences on the performance of the asphalt pavement. However, if the early cracks are not repaired in time, a seepage channel will be formed inside of the pavement structure, and water will penetrate the subgrade structure along with the seepage paths, leading to the deterioration of the underlying layers and the destruction of the stability of the whole structure.

A super absorbent polymer (SAP) is a kind of polymer material with a strong water absorption and water retention property. SAPs can absorb water with hundreds of times of its own mass [8]. After water absorption, the SAP powder becomes the SAP hydrogel, which features a certain viscosity, absorption, and strong plasticity [9]. The formation of the SAP hydrogel is amorphous, and it could release water gradually under a certain differential ion concentration and pressure. Therefore, SAP has been widely used in cementitious materials as an internal curing admixture and in soils as a water-retaining material [10,11,12,13]. During the fabrication of cement-based materials, SAPs could absorb and retain a large amount of water. The water inside of the material was consumed with the hydration process of cementitious materials, and the relative humidity in the microstructure of the material decreases rapidly, especially in its early age. Under these circumstances, SAP has the potential to release water to the surrounding when the moisture concentration outside is lower than that inside of the SAP hydrogel. In addition, for the un-hydrated cement, the absorbed water inside could help to promote the degree of hydration [14]. Studies showed that the water released from the SAP hydrogels was uniform and instantaneous at the early age of concrete, while the depercolation of the capillary porosity may substantially refrain the water migration at the late age [15]. Because of the water released from the SAP hydrogels, SAPs could enhance the cement hydration even at the later stage. Meanwhile, the SAP’s volume swells rapidly after water absorption [10,16]. If cracks are generated in the concrete, the swollen SAP hydrogels could clog the cracks leading to the benefit for crack filling [17]. Lee et al. [18] studied the optimal width of the SAP to block cracks, and it revealed that the SAP had the best sealing effect after re-expansion when the crack width was below 0.3 mm. Mignon et al. [19] found that SAPs could accelerate the formation of CaCO_3_ in the concrete and produce the main product of self-healing in cracks. It pointed out that the smaller the width of the crack, the higher the healing efficiency of the SAP. Deng et al. [20] have fabricated pre-cracked engineered cementitious composites (ECC) and proposed that SAPs could still absorb water to facilitate the self-healing of cracks under the condition of moisture deficiency, but the ECC incorporated with 4% SAP exhibited superior self-healing performance under the curing condition of 95% relative humidity. Chindasiriphan et al. [21] reported that the coupling effects of fly ash and SAP could effectively improve the self-healing ability of concrete, reaching 100% crack closure condition after 28 days of self-healing.

Although the development and application of SAPs in cement-based materials are widespread, the addition of SAP in asphaltic materials is currently insufficiently explored. A few studies have shown that SAP can be used as a warm mix additive in asphalt mixtures [22,23]. Two types of SAPs were considered in Mou [22] as warm mix additives, and a series of experiments were conducted to evaluate the effectiveness of the SAP in the study. The results demonstrated that the SAP could reduce the mixing and compaction temperature up to 20 °C, thus significantly saving the energy for production and compaction. Yuan [23] has compared the SAP warm mix additive (named Wsap) with the commonly used commercial warm mix additives, such as Sasibot, Aspha-min, and Evotherm, and it found that the Wsap can significantly improve the high temperature stability, low temperature stability, and rutting resistance of the asphalt concrete, and has no obvious effect on its fatigue resistance. The research from Li [24] indicated that an additional SAP in the asphalt mixture with large pores could effectively enhance its water-retaining capacity. After being modified with SAP, the strength and low-temperature performance of the large-pore asphalt mixture were both improved, but, meanwhile, its water stability was not affected, and its anti-skid performance was reduced.

In asphalt pavements, cracking is also a major distress under the coupling effect of traffic loading and environment factors [25]. The amorphous hydrogels formed by the swollen SAP after water absorption has the potential to fill the cracks in asphalt mixtures and thus reduce the infiltration of rainwater. Therefore, the application of SAP in asphalt pavements could endow the material with crack sealing and filling ability and prevent the continuous water migration along the crack openings. Moisture induced damages, including the pumping and adhesion failure at the binder-aggregate interface, are considered as major contributors to the deteriorations of the durability of asphalt pavements [26,27,28,29]. Moreover, under freeze-thaw attacks, the water trapped in the cracks and pores would produce considerable frost pressures, which could further aggravate the deterioration of asphalt concrete. Geng et al. [30] reported that the moisture susceptibility, rutting resistance, and low temperature cracking resistance of asphalt mixtures were improved when SAPs were introduced in the materials. The hemp fiber, needle-like zinc oxide whiskers, and SAP were also added into asphalt mixtures in Gu et al. [31], and the performances of the asphalt mixtures were investigated. The findings from this research revealed, that when rainwater infiltrated into asphalt concrete, the water could be stored by the SAP. When the pavement temperature rose, water could flow along the hemp fiber to the surface of the asphalt concrete and evaporated through the capillary action, which reduced the pavement temperature and improved the anti-cracking performance of the surface layer.

At present, a lot of research has been addressed on the application of SAP in cementitious materials, including the effectiveness of SAP as an internal curing admixture, the influence of SAP on the strength development of cement concrete, the reduction effect of SAP on shrinkage and hydration heat of cement concrete, and the influence on the durability of cement concrete, etc. However, its feasibility and applicability in asphalt materials still needs to be further discussed. After the SAP’s contact with water, especially for those in the crack opening path, it swells and forms amorphous hydrogels with a certain viscosity, which could obstruct the cracks and seepage channels, and therefore impede the water from migrating and infiltrating in the material. This function implies a benefit of utilizing SAP in asphalt pavements to resist hydrodynamic damages and potentially mitigate the water erosion in the underlying layers of the pavement structure. 

## 2. Objective and Scope

The objective of this study was to evaluate the feasibility and applicability of utilizing SAP in asphalt mixtures in an attempt to endow the pavement with a function of crack filling and infiltration obstruction. To achieve this goal, the fundamental physical and mechanical properties, rheological properties, and specially designed permeability tests for the asphalt mastics and asphalt mixtures with various contents of SAP were conducted. These properties are the prerequisites for SAPs’ application in asphalt pavements to fulfill its potential functions. Furthermore, the influence of SAP on the compatibility of the asphalt mixtures and the mechanisms for its functions in the asphalt pavements were also discussed. 

## 3. Materials and Test Methods

### 3.1. Materials

The 70# base asphalt produced by Maoming Sinopec (Maoming, China) that is commonly used in China was selected for the study. The basic properties of the asphalt are presented in Table 1.

The main components of the SAP used for the study are made of low crosslinking sodium polyacrylate, water, and a crosslinking agent. The production of this type of SAP consists of four main steps: (1) the reaction of acrylic acid (AA) and NaOH; (2) adding some cross-linking agents into the solution obtained in the first step; (3) the solution formed in the second step was subjected to a water bath reaction under a relative high temperature, e.g., 70 °C; and (4) drying and grinding. Some crucial properties of the SAP used for the study are shown in Table 2.

In addition, the hydrated lime was added into the asphalt mixtures as an anti-stripping additive in this study. Its density is 2.70 g/cm^3^, and the particle size is lower than 0.6 mm.

### 3.2. Sample Preparation and Mixture Design

#### 3.2.1. Preparation of SAP Modified Asphalt Mastic

The ratio of filler/asphalt was controlled at 1.0 in the test. Due to the significant difference in density between the SAP and mineral powder, the SAP was used to partially replace the mineral powder by means of the equivalent volume method. The preparation process of the SAP modified asphalt mastic was illustrated in Figure 1: (1) the asphalt binder was heated to 165 °C; (2) the required amount of mineral powder was added into the asphalt and stirred with a glass rod for 5 min; (3) at 165 °C, the mixture was blended with a high-speed shearing machine at 4500 rpm for 20 min; (4) the SAP was then added into the mixture with a mixing of 10 min at 1000 rpm and 20 min at 2500 rpm. 

#### 3.2.2. Mixture Design of the Asphalt Mixtures Incorporated with SAP

The synthetic gradation of the asphalt mixture (AC-13) is shown in Table 3. The optimum asphalt-aggregate ratio was determined to be 5.0%. In order to simplify the design process, only a certain content of the mineral powder was replaced by equivalent volume of SAP when the asphalt mixture was designed. The density and porosity of the specimens are shown in Table 4.

### 3.3. Testing Methods

#### 3.3.1. Basic Properties of SAP

(1) Water absorption rate

As shown in Figure 2, the mesh bag method was employed in the study to measure the water absorption rate of the SAP. The mesh bag was made of a 500-mesh nylon fabric. Prior to testing, the mesh bag was completely moistened. During the test, a moderate amount of SAP was capsuled in the mesh bag and immersed in the water for 2 min. After flipping and agitation, the mesh bag with the saturated SAP (SAP hydrogel) was then taken out and suspended in a shelf to filter out the excess water in the bag. The water absorption ratio *Q* for the saturated SAP was calculated with the following equation [24]:(1)$$Q=\frac{S_{z}-S_{1}-S_{0}}{S_{0}}$$
where Q is the water absorption ratio of SAP, mL/g; $S_{z}$ is the total mass of the saturated SAP and soaked mesh bag after fully absorbing water, g; $S_{1}$ is the wetted mesh bag mass, g; $S_{0}$ is the dry mass of SAP in the mesh bag, g.

The tests were carried out three times in parallel and the average water absorption ratio was obtained. The relationship between the water absorption time and water absorption ratio was established, and the slope of the tangent line at any point on the relation curve was the water absorption rate at that time.

(2) Water retention and release rate of the SAP

The water retention rate refers to the ability of the saturated SAP to store water under different environments. The water release rate is defined as the slope of the water retention ability, which reflects how fast the SAP releases water under different environments. The water retention rate and water release rate were examined under natural conditions for 14 days, or under 20 °C, 40 °C, 60 °C, and 80 °C oven drying conditions for 12 h. The water retention rate can be calculated with the following equation [24]:(2)$$\varphi =\frac{M_{t}-M_{0}-M_{2}}{M_{1}-M_{0}-M_{2}}\times 100\%$$
where φ is the water retention rate of the SAP, %; $M_{2}$ is the mass of dry SAP, g; $M_{0}$ is the mass of the dry beaker, g; $M_{1}$ is the total mass of the beaker and the SAP after water absorption, g; $M_{t}$ is the total mass of the beaker and the SAP after water absorption with various immersion durations.

(3) Cyclic water absorption capacity of the SAP

The cyclic water absorption capacity is the residual water absorption capacity of the SAP after repeated water absorption and drying. An oven was used to repeatedly dry the SAP after water absorption. The calculation process is shown in Equations (3) and (4) [24].

The water absorption rate for the first cycle is:(3)$$Q_{1}=\frac{M_{n1}-M_{1}-M_{2}}{M_{2}}\times 100\%$$

The loss of water absorption rate after several water absorbing and drying cycles can be calculated as:(4)$$Q_{n}=\frac{M_{nn}-M_{1}-M_{2}}{Q_{1}}\times 100\%$$
where $M_{0}$ is the mass of mesh bag, g; $M_{1}$ is the mass of the wetted mesh bag, g; $M_{2}$ is the mass of the fully dried SAP, g; $M_{n1}$ is the total mass of the saturated SAP and the mesh bag after the first full water absorption, g; $M_{nn}$ is the total mass of the saturated SAP and the mesh bag after the *n*th cycle of water absorption, g; $Q_{1}$ is the water absorption of the SAP at the first cycle, %; $Q_{n}$ is the loss of water absorption rate after the n^th^ cycle of water absorbing and drying, %.

#### 3.3.2. Morphology of the SAP and Mineral Powder

The shape, angularity, and surface roughness of the filler powder would directly affect the performance of the asphalt mixture. A JSM-IT500LV scanning electron microscope (JEOL Ltd, Akishima-shi, Japan) was used to observe the microstructure of the mineral powder and SAP, and as the mineral powder and SAP were wrapped in asphalt, the profiles of asphalt specimens were observed.

#### 3.3.3. Rheological Properties Test

The Anton Paar SmartPave102 dynamic shear rheometer (Anton Paar, Graz, Austria) was used to carry out the temperature sweep test, frequency sweep test, and linear amplitude sweep (LAS) test on the asphalt mastics with 0, 10%, 20%, 30%, and 40% of the mineral powder replaced by the SAP.

(1) Temperature sweep test

By continuous temperature change, temperature sweeping test can observe the changes of δ and $G^{*}/sin\delta$ of the asphalt mastic. The temperature of the test was set within the range of 40–90 °C, the heating rate was set to 2 °C/min. The strain control mode was adopted for the test, the strain and the frequency were controlled at 12% and 10 rad/s, respectively. The diameter of the specimens was 25 mm and the plate clearance was 1 mm.

(2) Frequency sweep test

The viscoelastic properties of the asphalt mastic varied with the frequency of loading. Therefore, the continuous frequency variation of the asphalt mastic can reflect the viscoelastic changes in the actual pavement. In this study, the temperatures of 46 °C, 52 °C, 58 °C, 64 °C, 70 °C, and 76 °C were selected for the frequency sweeping, and the applied frequency ranged from 0.1 Hz to 100 Hz for each temperature. The diameter of the specimens is 25 mm and the plate clearance is 1 mm.

(3) Linear amplitude sweep (LAS) test

The LAS test is based on viscoelastic continuum damage mechanics (VECD) to evaluate the fatigue damage characteristics of the asphalt mastic under repeated loads. According to the AASHTO TP 101 specification [32], the LAS test consists of two parts: frequency sweep and amplitude sweep. Firstly, the frequency sweeping was carried out in the test with a strain level at 0.1% and a frequency from 0.2–30 Hz. The relation curve between the storage modulus (G’) and frequency of the tested samples was obtained. After the sample was relaxed for 2 min, the amplitude sweeping with a strain level from 0.1–30% was then performed at 10 Hz to evaluate the fatigue damage characteristics of the asphalt mastic. The test temperature was 25 °C, the diameter of the specimens was 8 mm, and the plate clearance was 2 mm.

The anti-fatigue strength was calculated according to the results of the frequency and amplitude sweeping, as shown in the Equations (5)–(8) [32]. 

(1) The cumulative damage D(t) of the specimen can be expressed as:(5)$$D\left(t\right)≅\sum_{i=1}^{N}{\left[\pi I_{D}\gamma_{0}^{2}\left(\left|G^{*}\right|sin\delta_{i-1}-\left|G^{*}\right|sin\delta_{i}\right)\right]}^{\frac{\alpha }{1+\alpha }}{\left(t_{i}-t_{i-1}\right)}^{\frac{1}{1+\alpha }}$$
where $I_{D}$ is the complex shear modulus at 1% strain level, MPa; $\gamma_{0}$ is the strain of data points; t is the test time, s; α is the material parameter in the non-damaged state, $\alpha =1+\frac{1}{m}$.

(2) At any time, *t*, the $\left|G^{*}\right|sin\delta$, and D(t) can be transformed into a power function and then fitted with the following equation:(6)$$\left|G^{*}\right|sin\delta =C_{0}-C_{1}{\left(D\right)}^{C_{2}}$$
where $C_{0}$ is the value of $\left|G^{*}\right|sin\delta$ when the strain is 0.1%; $C_{1}$ and $C_{2}$ are curve fitting parameters that can be obtained by the equation: $\log\left(C_{0}-\left|G^{*}\right|sin\delta \right)=\log\left(C_{1}\right)+C_{2}\cdot \log\left(D\right)$.

(3) The relevant parameter $A_{35}$ is required to calculate the fatigue life:(7)$$A_{35}=\frac{f{\left(D_{f}\right)}^{k}}{k{\left(\pi I_{D}C_{1}C_{2}\right)}^{\alpha }}$$
where f is loading frequency, 10 Hz; $k=1+\left(1-C_{2}\right)\alpha$; B=2α. 

(4) The fatigue life can be calculated by the following formula:(8)$$N_{f}=A_{35}{\left(\gamma_{\max}\right)}^{-B}$$
where $\gamma_{max}$ is the expected maximum strain of the asphalt pavement structure.

#### 3.3.4. High Temperature Stability Test

The Marshall stability test and the rutting performance test were carried out to evaluate the high temperature stability of asphalt mixtures incorporated with various contents of SAP.

#### 3.3.5. Water Stability Test

SAP would absorb water in the process of use, which may affect the adhesion performance between asphalt and aggregate. Therefore, it is necessary to evaluate the water stability of the asphalt mixtures incorporated with the SAP, and the immersion Marshall stability test, freeze-thaw splitting strength test, and Cantabro test were carried out for this purpose. The water stability of the SAP modified asphalt mixtures was evaluated by the Marshall stability ($MS_{0}$), tensile strength ratio (TSR) after freeze-thaw cycles, and the weight loss (ΔS) of the asphalt mixtures.

#### 3.3.6. Permeability Test

The device shown in Figure 3 was specially designed to evaluate the ability of SAPs on reducing the water migration and infiltration in asphalt mixtures. The downpipe is made of a transparent acrylic pipe with 1000 mm in length, 100 mm in diameter, and 5 mm in thickness. During the test, a flexible heat shrinkable tube made of polyolefin was used to wrap the specimen tightly and guarantee that the water could only drain down through the section of the specimen. The drainpipe is a PVC pipe with 40 mm in diameter and 3.2 mm in thickness. The specimens of the asphalt mixture used for this test were fabricated with the standard Marshall method, and their diameter and height were 101.6 mm and 63.5 ± 1.2 mm, respectively.

In this method, the heat shrinkable tube was used to seal the specimen in its position, and the change of permeability of the specimen under the action of water pressure was used to characterize the effect of the SAP on the reduction in water migration and infiltration. Specifically, the curve between the hydraulic gradient, *i*, and the discharge velocity, *v*, were drawn by recording the scale on the PVC pipe, and the permeability coefficient, *K*, was calculated with the Equations (9)–(13) [33]. 

The relationship between the height of the water head in the downpipe and time duration of the test can be fitted by a quadratic equation as:(9)$$h=a_{0}+a_{1}t+a_{2}t^{2}$$
where $a_{0}$, $a_{1}$, and $a_{2}$ are the regression coefficients; h is height of water head in the downpipe, mm; t is the time duration, s.

The following differential equation can be obtained by taking the derivative of the above equation:(10)$$\frac{dh}{dt}=\alpha_{1}+\alpha_{2}t$$
where $\alpha_{1}$ and $\alpha_{2}$ are the regression coefficients of differential equations of the head pressure and time.

Therefore, the discharging rate of water head can be expressed as:(11)$$v=\frac{dQ}{dt}=\frac{A_{1}}{A_{2}}\frac{dh}{dt}=\frac{r_{1}^{2}}{r_{2}^{2}}\frac{dh}{dt}$$
where v is the discharging rate of water head, mm/s; Q is the total amount of water that passes through the specimen, ml; $A_{1}$ and $A_{2}$ are the cross-sectional areas of the downpipe and specimen, respectively. $r_{1}$ and $r_{2}$ are the radii of the downpipe and specimen, respectively.

The hydraulic gradient can be expressed as follows:(12)$$i=\frac{\Delta h}{l}$$
where i is the hydraulic gradient; Δh is the difference between the inlet and outlet water head that flows through the specimen, mm; l is the height of the specimen, mm.

Finally, the permeability coefficient, K, with the variable water head can be expressed as:(13)$$K=2.3\frac{A_{1}l}{A_{2}\left(t_{2}-t_{1}\right)}ln\frac{h_{1}}{h_{2}}$$
where, K is the permeability coefficient of the specimen under various water heads; $t_{1}$ and $t_{2}$ are the starting and ending time for the measurement of the water head, respectively, s; $h_{1}$ and $h_{2}$ are the starting and ending height of the water head, respectively, mm.

## 4. Test Results and Analysis

### 4.1. Basic Properties of the SAP

(1) Water absorption ratio and rate

As can be seen from Figure 4, the water absorption ratio of the SAP was basically unchanged after the immersion in water for 15 s. It could be considered that the maximum water absorption ratio was reached at this time, and the water absorption ratio was 105 times that of the mass of the dry SAP. The water absorption rate of the SAP reached its maximum value in the first 2 s, and decreased significantly in the following 2–5 s. The change of water absorption rate was no longer observable after 5 s, and the rate dropped to 0 and kept stable after 15 s, which indicates that the SAP achieved its maximum saturation. The greater the water absorption ratio of the SAP, the more rainwater it could absorb, and the greater the water absorption rate in the early stage, the more obvious the instantaneous obstruction could be achieved for rainwater infiltration.

(2) Water retention and release rate

The water retention and release rate of the SAP are illustrated in Figure 5. The water retention rate could be maintained above 80% on the first day under room temperature of 20 ± 2 °C. The water release rate of SAP reached its maximum value around 24 h, which was 20.94 g/d. From the next day, the water release rate tended to decline gradually, and its final value under the room temperature on the 14th day was around 12%.

As the natural state was affected by the temperature, humidity, wind speed, and other factors, the water retention rate and water release rate at different temperatures under oven drying conditions were further discussed, as shown in Figure 6. With the increase in temperature, the water retention rate of the SAP decreased significantly after 14 days, from 89.58% at 20 °C to 1.73% at 80 °C. The results indicated that a high temperature would destroy the internal network structure of the SAP, and the increase in the energy of water molecules would make the water easier to escape from the network structure. Regarding the water release rate, the higher the temperature was, the faster the water release rate would be. Meanwhile, it could be found that the water release rate for the specimens under various temperatures all reached its maximum value on the first day, and the water release rate decreased with the increase in time. This indicated that the SAP exhibited stronger water retention ability when its internal water was less. 

(3) Cyclic water absorption and release rate

Due to the repeated absorption-evaporation cycle of the pavement surface, the SAP should have excellent circulating water absorption and release capacity, so that it can perform its function repeatedly. As can be seen from Figure 7, the water absorption capacity of the SAP decreased significantly (about 6.67%) from the first cycle to the second cycle. From the second to the fifth cycle, the attenuation was relatively gentle, and the attenuation for a single cycle remained within 4%. After five cycles, the SAP still maintained 82.86% water absorption capacity, compared with its initial state.

### 4.2. Microscopic Morphology of SAP and Mineral Powder

The microscopic morphology of the SAP and mineral powder are displayed in Figure 8. The results from SEM showed that the SAP had a similar appearance to the mineral powder, and they were both irregular granular, but the SAP are relatively finer than the mineral powder and easier to clump together. The surface of the mineral powder was rougher than that of the SAP particles, which could increase their specific surface area and the adhesion performance to the asphalt binder.

According to Figure 9, both the mineral powder and SAP could be dispersed uniformly in the asphalt binder, but it was found that the mineral powder particles were completely wrapped in the asphalt. This was because the mineral powder was alkaline and had higher surface energy, which enables a better blending effect and adhesion between mineral powder particles and asphalt binder. Due to the relatively low alkalinity of the SAP, the adhesion between the SAP and asphalt was weaker than that of the mineral powder. However, it is still visible that a part of the exposed SAP particles was embedded in the asphalt binder, while there was a small amount of asphalt wrapping on the exposed surface of the SAP. This is due to the rough surface of the SAP particles, which enhances the adhesion between the SAP and asphalt to a certain extent.

### 4.3. Rheological Properties of SAP Modified Asphalt Mastic

(1) Temperature sweep test

The changes of δ and $G^{*}/sin\delta$ of asphalt mastics with 0%, 10%, 20%, 30%, and 40% content of SAP were shown in Figure 10. At the same temperature, the $G^{*}/sin\delta$ increased when the content of the SAP is less than 10% but decreased when the content of the SAP is greater than 10%. When the temperature was 65 °C, the $G^{*}/sin\delta$ increased by 0.7% when the content of SAP was 10%, but decreased by 6.3%, 13.3%, and 14.7% when the content of SAP was 20%, 30%, and 40%, respectively. This indicates that the increase in the SAP content would weaken the high temperature performance of the asphalt mastics. In the asphalt mastics, the SAP is easier to agglomerate than the mineral powder at high temperatures, which could impair its high temperature stability. Moreover, it can also be seen that there were no remarkable changes which could be observed on the results of the phase angle δ, indicating that the SAP exhibited insignificant effects on the viscoelastic state of the asphalt mastics.

(2) Frequency sweep test

At present, there are many equations derived for the time-temperature equivalence principle, and the WLF equation proposed by M.L. Williams, R.F. Landel, and J.D. Ferry in the 1960s is the most widely used one for viscoelastic asphalt mastics [34]. When the test temperature, *t*, and the glass transition temperature, *T_g_,* satisfies the inequality: *T_g_ < t < T_g_ + 100*, the displacement factor $\alpha_{T}$ can be obtained through the following formula:(14)$$\mathrm{lg}\left(\alpha_{T}\right)=\frac{-C_{1}\left(T-T_{0}\right)}{C_{2}+\left(T-T_{0}\right)}$$
where $\alpha_{T}$ is the displacement factor; T is the test temperature, °C; $T_{0}$ is the reference temperature, °C; $C_{1}$ and $C_{2}$ are the empirical parameters.

According to the principle of time-temperature equivalence, the master curve of the complex shear modulus $G^{*}$ of the asphalt mastic was obtained. Furthermore, 58 °C was selected as the reference temperature for the master curve. As shown in Figure 11, with the increase in the loading frequency, the $G^{*}$ of the asphalt mastic increased, indicating that the high temperature performance of the asphalt mastic increased with the increase in loading frequency. The frequency mainly reflects the duration of the traffic load on the asphalt pavement, that is, the higher the frequency, the shorter the loading time. The $G^{*}$ of all asphalt mastics decreased with the increase in the temperature. At the same temperature, when the content of SAP was 0%, 10%, 20%, 30%, and 40%, the $G^{*}$ of asphalt mastic showed a trend of increasing in the beginning and then decreased and reached the maximum value when the content of SAP was between 10% to 20%. Therefore, the replacement of the mineral powder with more than 10% SAP could adversely affect the high temperature performance of the asphalt mastic, which was consistent with the conclusion from the temperature sweep tests in the previous section.

In the low frequency region, the viscoelastic properties of all specimens were very close to each other. In the high frequency region, the viscoelastic properties of the specimens varied distinctly. This indicated that the SAP demonstrated a significant impact on the low temperature performance of the asphalt mastics. 

(3) Linear amplitude sweep (LAS) test

The fatigue life of the asphalt mastic was evaluated with the LAS tests, in accordance with AASHTO TP 101. Firstly, a frequency sweeping was performed to determine the material parameter, α, for the asphalt mastic in the non-damaged state by fitting the relationship between the storage modulus and frequency. The fitted curve is shown in Figure 12.

The fitting results of the storage modulus (G ’) and the sweeping frequency (w) under different SAP replacement proportions are shown in Table 5:

After the frequency sweeping, the specimens were allowed to relax for 2 min, and then the amplitude sweeping was carried out on them to determine the fatigue damage parameters of the asphalt mastics. As shown in Figure 13, with the increase in strain, the stress increased first and then decreased. The initial stage of the development of stress can be regarded as a linear evolution. Before reaching the maximum stress, the stress of the specimen is proportional to the strain, that is, the asphalt mastic is considered in an elastic deformation state, and the slope reflects its resistance to the deformation. With the increased content of the SAP, the slope of the curve decreased, and the ability of the asphalt mastic to resist deformation was weakened. When the proportion of the SAP replacing the mineral powder increased from 0% to 40%, the strain reached the maximum value at 5.79%, 6.53%, 6.40%, 6.31%, and 6.27%, respectively, and the maximum stress at that time was 0.254 MPa, 0.255 MPa, 0.245 MPa, 0.235 MPa, and 0.225 MPa, respectively. With the increasing content of SAP, the asphalt mastic became softer, which can be observed with the reduced slope of the results. Therefore, a greater strain was required in this case when the asphalt mastic reached its maximum stress. 

Fatigue damage parameters were calculated according to the Formulas (5) to (8), and the fatigue lives of the specimens at two typical strain levels (γ = 2.5% and γ = 5.0%) were analyzed. It can be seen from Figure 14, when the content of SAP was 10%, the fatigue life of the asphalt mastic increased by 25.2% at 2.5% strain and 7.0% at 5% strain compared with the asphalt mastic without the SAP. After the content of the SAP increased to 20%, the fatigue life decreased obviously, especially when the content of the SAP reached 40%, the fatigue life is only 27.1% of that for the asphalt mastic without the SAP at 2.5% strain, and only 22.8% of that for the asphalt mastic without the SAP at 5% strain. This indicated that when the SAP replaces more than 10% of the mineral powder in the asphalt mastics, the higher the content, the greater the declination in the fatigue life of the asphalt. Adding too much SAP in the mixture could affect the adhesion between the SAP and the asphalt binder, thus making the asphalt mastic more prone to deform, and more likely to be damaged under repeated loads.

### 4.4. High Temperature Stability Test

(1) The Marshall stability test

According to Figure 15, the Marshall stability of all asphalt mixtures with various contents of the SAP met the requirement of no less than 8 kN, and the order of stability results was SAP-10% > SAP-0 > SAP-20% > SAP-30% > SAP-40%. When 10% of the mineral powder was replaced by the SAP, the SAP could better bond with the asphalt binder due to the smaller particle size of the SAP, thus increasing the Marshall stability of the mixture. However, when the replacement proportion exceeded 20%, the bonding strength between the SAP and asphalt binder was deteriorated, and the Marshall stability of the mixture was also minimized to a certain extent. Therefore, the replacement proportion of the SAP to the mineral powder should be controlled less than 20% based on the high temperature stability of the asphalt mixtures. 

(2) The rutting performance test

The rutting resistance is an important index to evaluate the deformation resistance of the asphalt mixture. As illustrated in Figure 16, the dynamic stability from rutting performance tests for the asphalt mixtures with and without the SAP all met the requirement of >1000 times/mm. However, the dynamic stability of the asphalt mixtures modified with the SAP decreased slightly with the increase content of the SAP, which was consistent with the results from the Marshall stability tests.

### 4.5. Water Stability Test

Prior to the water stability tests, the appearance of the SAP modified asphalt mixtures under a long-term water immersion condition were analyzed. In order to explore the influence of the SAP on the stability of the asphalt mixture under the most unfavorable conditions, the specimens with 40% content of the SAP were prepared and submerged in a water bath at 25 °C for 90 days. According to Figure 17, after the immersion of 90 days, the change on the apparent characteristics of the asphalt mixtures modified with the SAP was inapparent, and no raveling and surface expansion could be observed during the process, only a part of SAP hydrogels could be seen on the surface of the specimen. 

Figure 18 shows the results from the immersion Marshall tests. The retained stability of the asphalt mixtures modified with the SAP met the requirements of the specification ≥80%. However, with the increased proportion of the SAP, the retained stability decreased gradually. Compared with the asphalt mixtures without the SAP, the retained stability decreased by 2.71%, 5.02%, 7.56%, and 11.15%, respectively. According to the analysis, the hydrogel formed by the SAP inside of the asphalt mixture may play a lubricating role in the interface between the aggregates, thus weakening the interlocking actions between the aggregate particles. Therefore, the mineral powder partially replaced by the SAP showed negative impacts on the water stability of the asphalt mixture.

According to the results shown in Figure 19, the TSR obtained from the freeze-thaw splitting test decreased with the increased content of the SAP. During the freeze-thaw cycle, the exposed SAP in the asphalt mixture could absorb a large amount of water and become amorphous hydrogels filling in the pores. When the temperature was below 0 °C, the hydrogels in the pores freeze and generate expansion stress to the matrix. The repeated frost forces caused by the hydrogels could induce internal damages to the asphalt mixture. 

It can be seen from Figure 20 that the weight loss from both the standard Cantabro tests and immersion Cantabro tests continued to increase with the increased content of the SAP, but both met the specification requirements of <20%. In the standard Cantabro test, when SAP replaced 10% mineral powder, the weight loss of the asphalt mixture did not increase significantly compared with the asphalt mixture without the SAP. However, in the immersion Cantabro test, after the immersion in a water bath for 20 h, the hydrogels were formed inside of the asphalt mixtures, which weakened the adhesion between the asphalt and aggregate. Therefore, the weight loss of the asphalt mixture after the immersion decreased slightly. When the SAP replacement ratio exceeded 10%, the weight losses of the mixtures were further aggravated with the poor adhesion between the asphalt binder and the SAP. In the immersion Cantabro test, the internal SAP also played a lubricating role in the mixture, so that the aggregate was more prone to fall off in the collision process. In consequence, the weight losses from the immersion Cantabro tests continued to increase with the increased proportion of SAP replacing mineral powder.

### 4.6. Permeability Test

(1) Permeability test results

As shown in Table 6, the height of the water head, h; the water head discharge rate, v; the slope, v’; of discharge rate of the water head; and the permeability coefficient, K, were obtained based on testing results. 

According to the results presented in Figure 21, as the increased content of SAP, the discharge velocity of water head became slower at the same hydraulic gradient i. This indicated that the more SAP incorporated in the asphalt mixture, the more obvious it could reduce the dynamic water pressure of the asphalt mixture under a certain hydraulic pressure. In the initial hydraulic gradient, compared with the asphalt mixture without the SAP, the initial water discharge rate decreased by 32.32%, 41.41%, 46.47%, and 51.52% with the increased SAP contents. By observing the v-i curve, with the decreased hydraulic head, the hydrodynamic pressure also decreased gradually, and the restrain effect of the SAP on the water infiltration became stronger and stronger, and this phenomenon became more significant with the decrease in the hydraulic gradient.

Figure 22 illustrates the permeability coefficient obtained from the tests. Reductions were discovered on the permeability coefficient of the asphalt mixtures with the SAP modified compared to the mixtures without the SAP. The permeability coefficient decreased by 34.83%, 44.43%, 49.82%, and 54.72%, respectively, when 10%, 20%, 30%, and 40% mineral powder was replaced by the SAP. The more the SAP incorporated in the asphalt mixture, the more significant effect of SAP on reducing the water infiltration could be observed.

(2) Mechanism on the reduction of infiltration 

Micro-cracks are inevitable in the service life of an asphalt pavement due to the actions of traffic loads and environmental media, and those micro-cracks would eventually become the channels for the infiltration of the rainwater. At the same time, the water trapped in the cracks would produce repeated hydrodynamic pressures under the action of traffic loads, which could further intensify the development of the cracks and water erosions in the pavement structure. Due to the addition of the SAP in the asphalt mixture, when the micro-cracks appear on the pavement, the SAP existing in the cracks would swell and fill the opening of the cracks after water absorption, as shown in Figure 23.

In order to better understand the presence of the SAP in the section of the specimen, the HiRox KH-7700 three-dimensional digital microscope, as shown in Figure 24a. was used to observe the sections of the asphalt mixture specimens with and without the SAP. The observation results showed that the SAP could be evenly distributed on the section of the asphalt specimens and absorb water rapidly and expand into hydrogel when the water invaded from the micro-cracks of the material. The expansion diameter in this case reached 1413.775 μm, which is sufficient to obstruct the opening of general micro-cracks in the early stage of the pavement cracking.

The function of the SAP can be interpreted by the schematic diagram of Figure 25. The SAP on both sides of the cracks can absorb water and form hydrogels, which can inhibit the flow of water in the fracture or crack. The SAP hydrogel is a soft and amorphous gel with a certain viscosity, which features a strong plasticity and would not produce extra internal forces to the asphalt mixture. When encountering obstacles, it will find other interspaces and eventually fill the micro-cracks or pores to ensure that water could not continue to penetrate and damage the structure. The SAP hydrogels cannot only clog the migration of moisture in the material under the water pressure but restrain the penetration of the rainwater in the asphalt pavement, and potentially mitigate the deterioration induced by the repeated hydrodynamic pressures on the pavement material and structure. On the other hand, the water in the SAP hydrogel will be gradually released after a period due to the concentration difference between the hydrogel and the atmosphere, and the SAP hydrogel will be slowly restored into a damp-dry granular material. This implies that the SAP can be repeatedly used in the pavement with its excellent circulating water absorption and release ability. 

## 5. Conclusions

The mechanical performances and functions of the asphalt mastics and mixtures modified with SAP were investigated with a series of experiments in this study. Based on the testing results, the conclusions can be drawn as follows:

(1) An appropriate addition of SAP could improve the rutting and fatigue resistances of the asphalt mastics. When 10% of the mineral powder was replaced by SAP, the high temperature performance of asphalt mastics could be slightly improved due to the better dispersion of the SAP. However, the SAP in the asphalt binder was easier to agglomerate than the mineral powder if its content was too large, which would cause an adverse effect on their high temperature performance. In terms of the fatigue performance at the medium temperature, the fatigue life of the asphalt mastic with 10% SAP increased by 25.2% at 2.5% strain and 7.0% at 5% strain compared with the asphalt mastic without SAP, but a reduction on the fatigue resistance of the asphalt mastic was observed after the content of SAP exceeded 20%.

(2) According to the results from water stability tests, all the asphalt mixtures could meet the specification requirements when the mineral powder was replaced by less than 30% of the SAP. The water stability of the asphalt mixtures decreased with the increasing contents of the SAP. After the immersion in the water, the SAP hydrogels formed inside of the asphalt mixtures could weaken the adhesion between the asphalt mastic and aggregates, thus impairing the water stability of the asphalt mixture. On the consideration of the fundamental performances, the proportion of the SAP replacing mineral powder in the asphalt mixture is better to be maintained within 20%.

(3) The SAP incorporated in the asphalt mixture could reduce its permeability and exhibited an observable reduction in the water infiltration of the asphalt concrete. The more the SAP was added in the mixture, the more significant the function could be achieved. Based on the analysis of the mesoscopic observation, the SAP distributed on both sides of the microfractures or microcracks could effectively impede the migration and infiltration of the rainwater in the material and thus has the potential to enhance the hydrodynamic susceptibility of the pavement structure.

## Figures and Tables

**Figure 1 materials-16-01082-f001:**
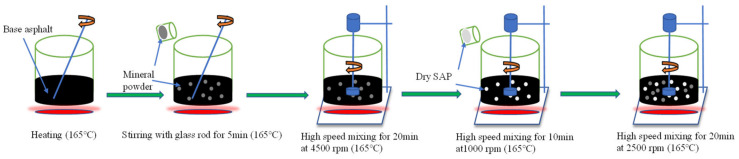
Preparation process of the SAP modified asphalt mastic.

**Figure 2 materials-16-01082-f002:**
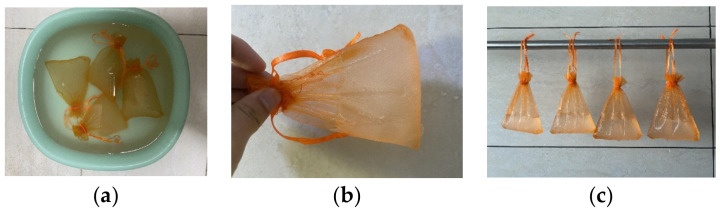
Water absorption test for SAP. (**a**) Immersion in water; (**b**) saturated SAP (SAP hydrogel) in the mesh bag; (**c**) filter out excess water in the mesh bag.

**Figure 3 materials-16-01082-f003:**
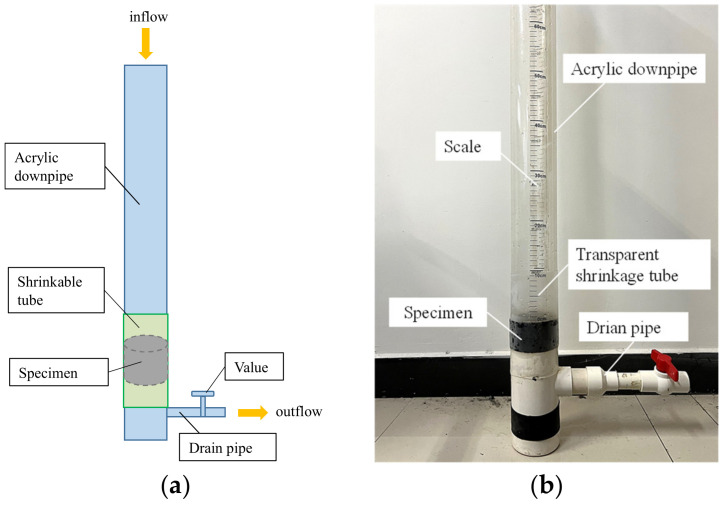
Permeability test device. (**a**) Schematic diagram of test setup; (**b**) Permeability test.

**Figure 4 materials-16-01082-f004:**
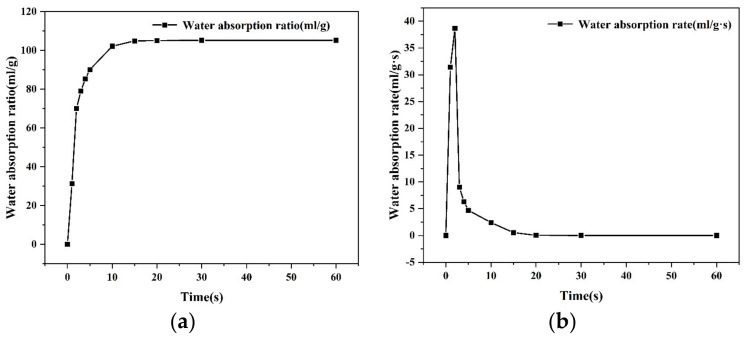
Water absorption ratio and rate of SAP. (**a**) Water absorption ratio; (**b**) water absorption rate.

**Figure 5 materials-16-01082-f005:**
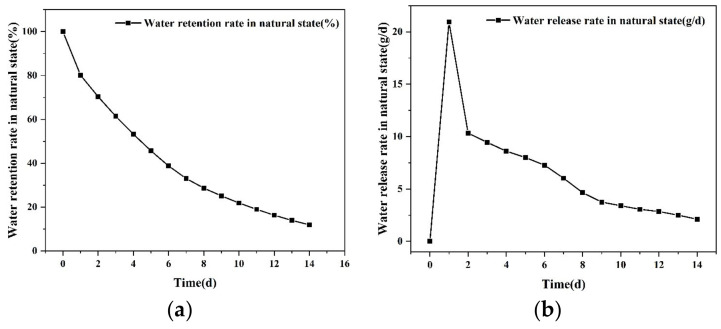
Water retention and release rate in natural state. (**a**) Water retention rate; (**b**) water release rate.

**Figure 6 materials-16-01082-f006:**
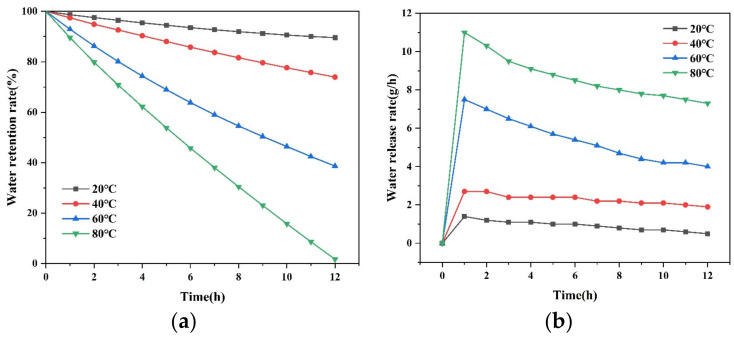
Water retention and release rate at different temperatures. (**a**) Water retention rate; (**b**) water release rate.

**Figure 7 materials-16-01082-f007:**
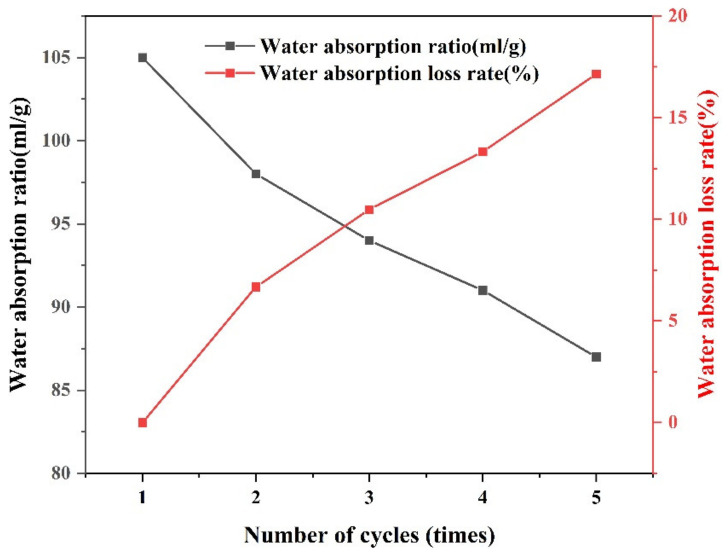
Cyclic water absorption rate.

**Figure 8 materials-16-01082-f008:**
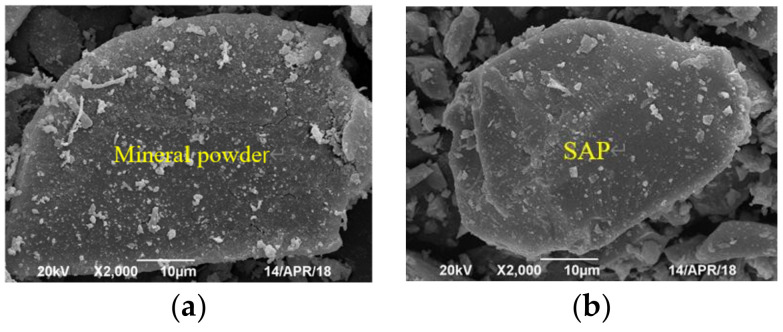
Microscopic morphology of mineral powder and SAP. (**a**) Mineral powder; (**b**) SAP.

**Figure 9 materials-16-01082-f009:**
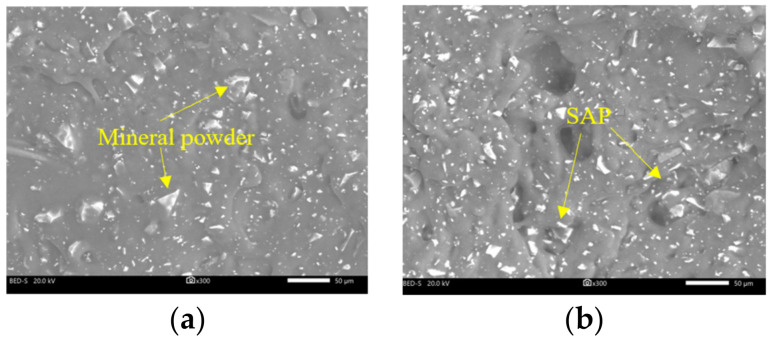
The state of mineral powder and SAP in asphalt binder. (**a**) Mineral powder; (**b**) SAP.

**Figure 10 materials-16-01082-f010:**
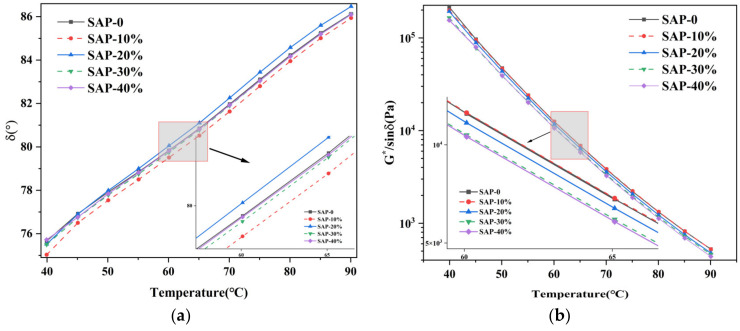
Results from temperature sweep tests. (**a**) δ; (**b**) $G^{*}/sin\delta$.

**Figure 11 materials-16-01082-f011:**
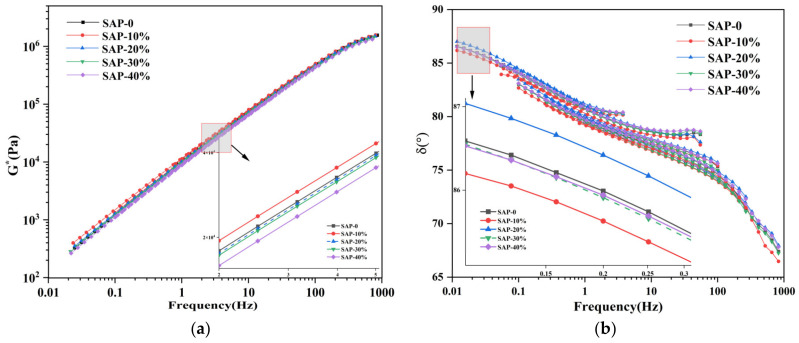
The result of frequency sweep: (**a**) $G^{*}$; (**b**) δ.

**Figure 12 materials-16-01082-f012:**
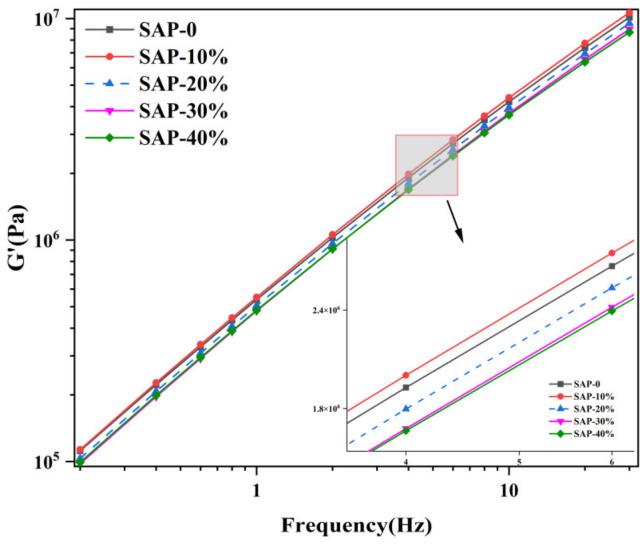
Storage modulus vs. frequency.

**Figure 13 materials-16-01082-f013:**
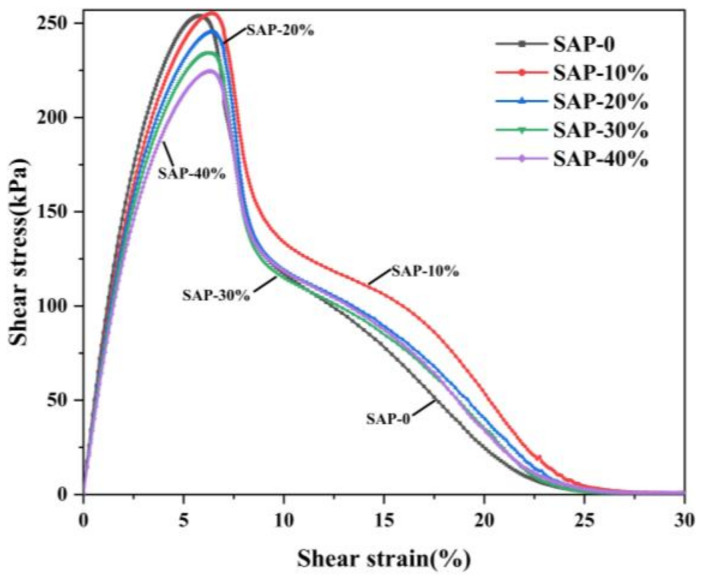
Results from amplitude sweep tests.

**Figure 14 materials-16-01082-f014:**
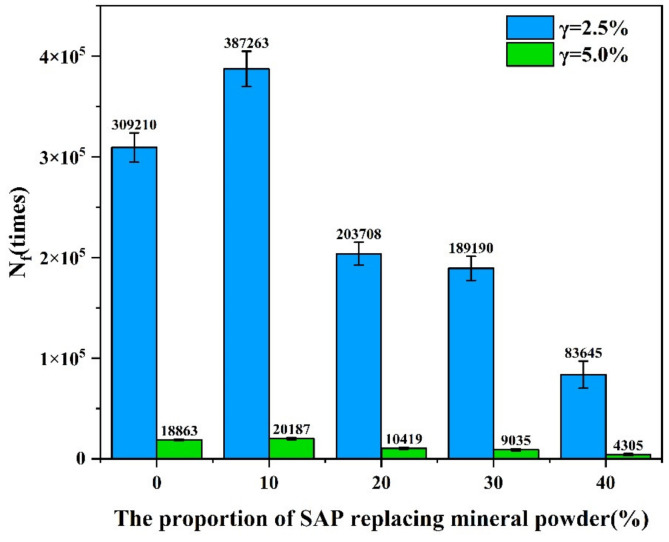
Fatigue life of asphalt mastic at 2.5% and 5% strain levels.

**Figure 15 materials-16-01082-f015:**
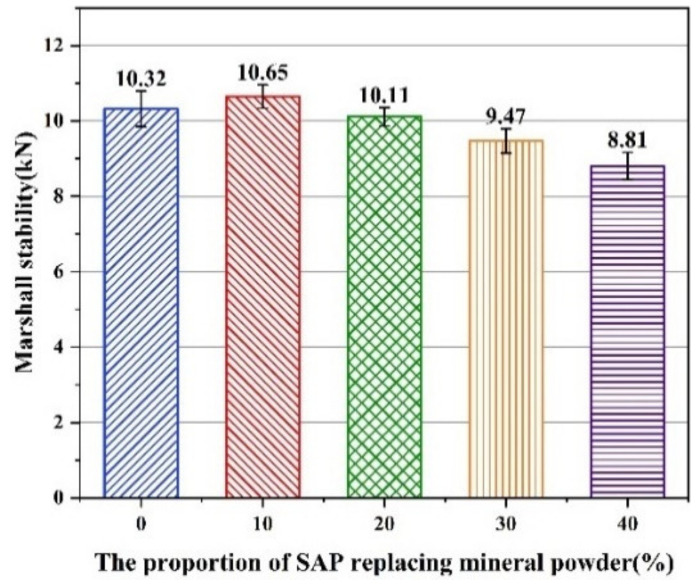
Results from Marshall stability test.

**Figure 16 materials-16-01082-f016:**
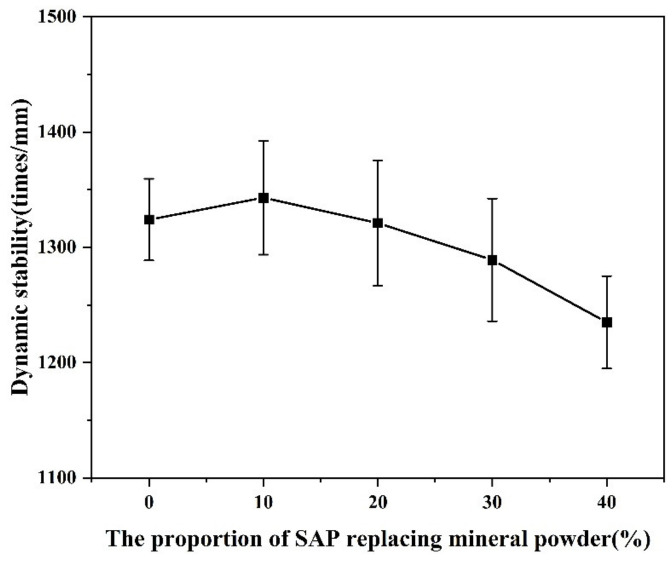
Results from rutting performance tests.

**Figure 17 materials-16-01082-f017:**
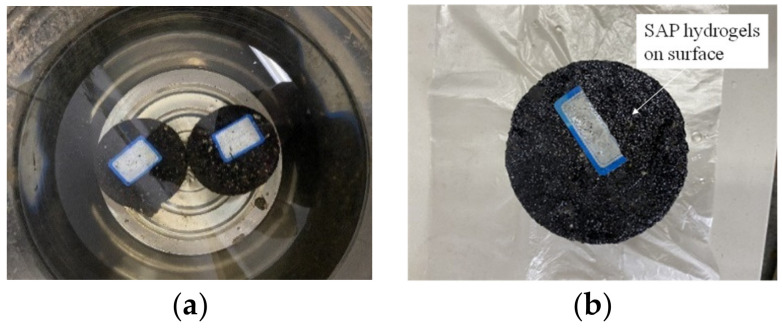
Long term immersion test. (**a**) Immersed specimen; (**b**) appearance after 90 days.

**Figure 18 materials-16-01082-f018:**
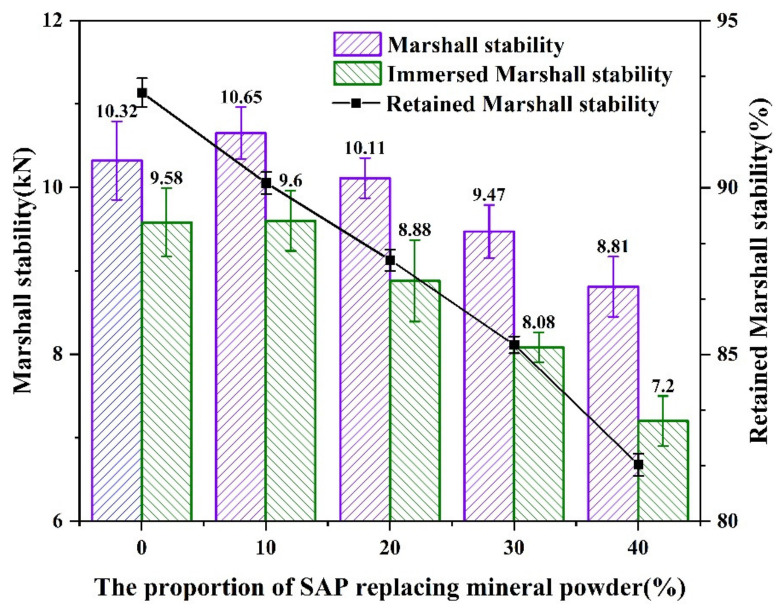
Result from immersion Marshall test.

**Figure 19 materials-16-01082-f019:**
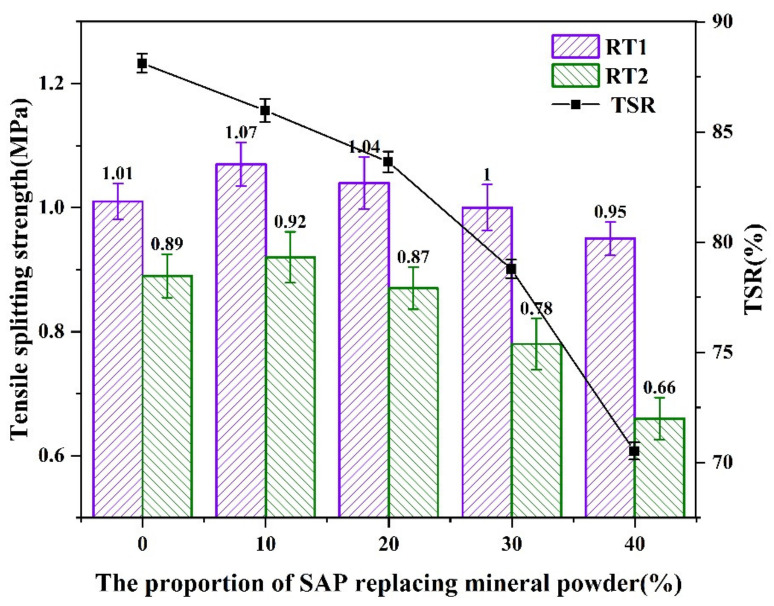
Result from freeze-thaw splitting tests.

**Figure 20 materials-16-01082-f020:**
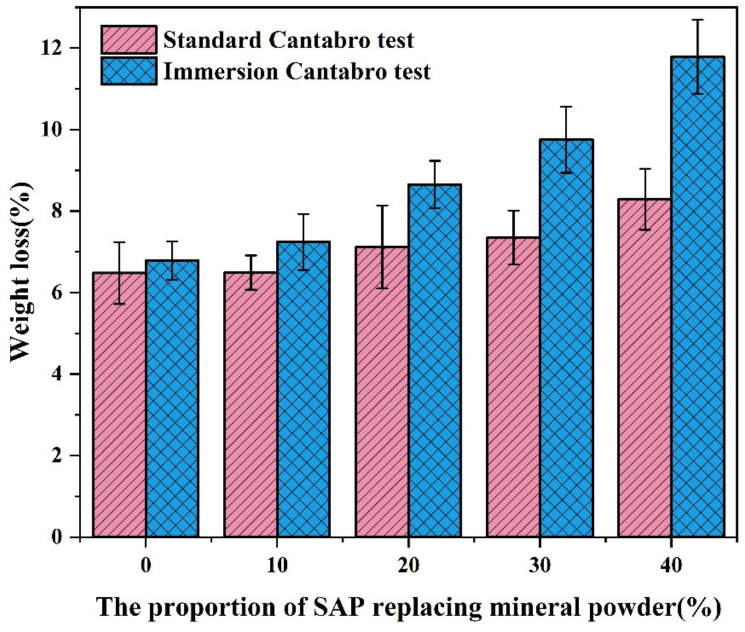
Results from Cantabro tests.

**Figure 21 materials-16-01082-f021:**
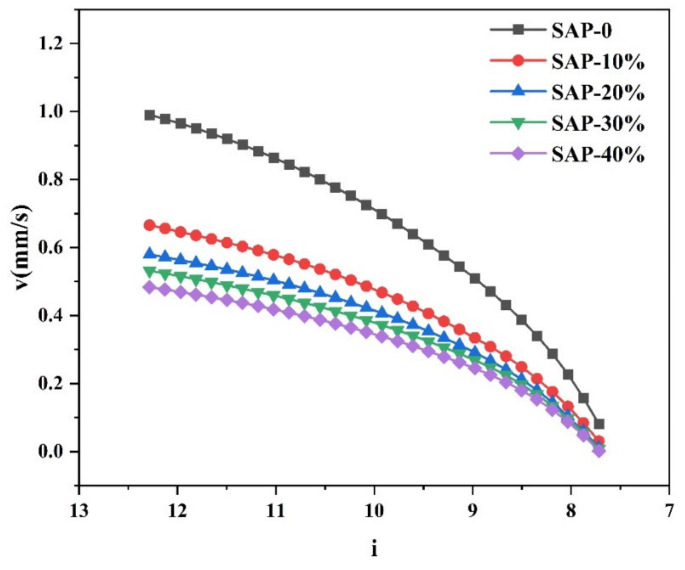
Results of water discharge velocity vs. hydraulic gradient.

**Figure 22 materials-16-01082-f022:**
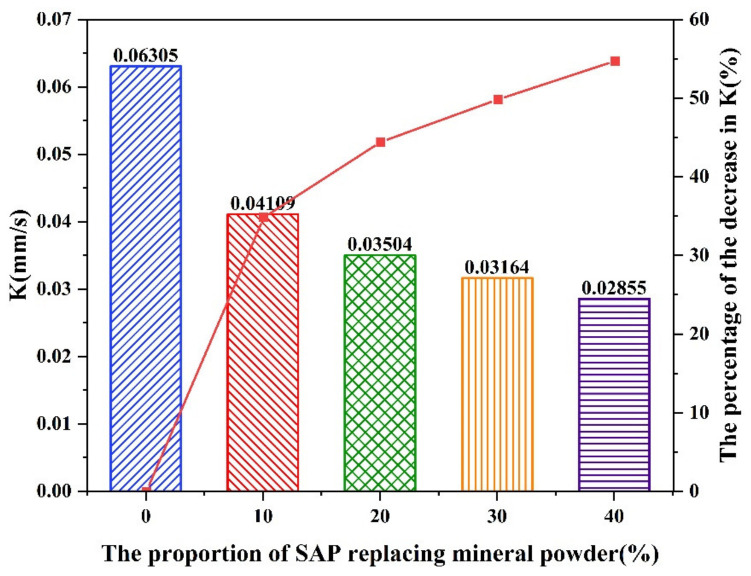
Results of permeability coefficient.

**Figure 23 materials-16-01082-f023:**
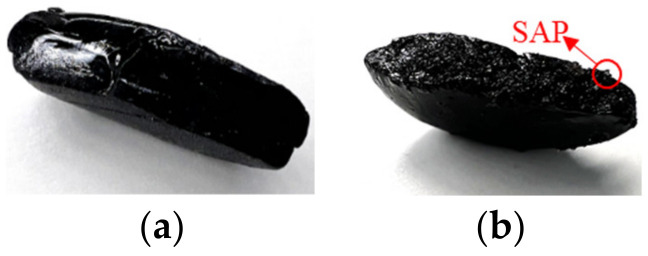
The sections of specimens: (**a**) the section of the specimen without SAP; (**b**) the section of the specimen with SAP.

**Figure 24 materials-16-01082-f024:**
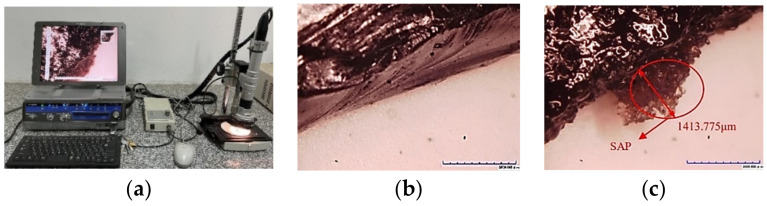
The 3D digital microscope and observation results: (**a**) three-dimensional digital microscope; (**b**) the section of the specimen without SAP; (**c**) the section of the specimen with SAP.

**Figure 25 materials-16-01082-f025:**
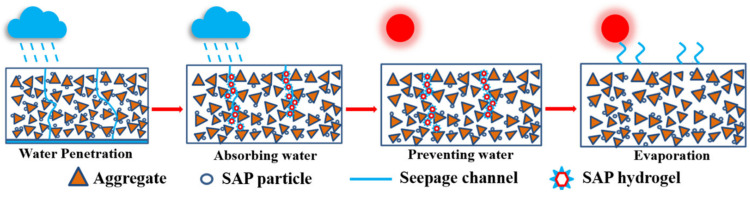
The function of SAP.

**Table 1 materials-16-01082-t001:** Basic properties of the asphalt used for study.

Properties	Technical Requirements	Test Results
Penetration (25 °C, 100 g, 5 s) (0.1 mm)	60~80	70
Softening point (Ring ball) (°C)	≥46	48.0
Ductility (5 cm/min, 10 °C) (cm)	≥15	26.4
Ductility (5 cm/min, 15 °C) (cm)	≥100	>100
Rolling Thin Film Oven Test (RTFOT)		
Mass loss (163 °C, 5 h) (%)	−0.8~+0.8	−0.056
Residual penetration ratio (163 °C, 5 h) (%)	≥61	64.3
Residual ductility (168 °C, 5 h) (cm)	≥6	6.4

**Table 2 materials-16-01082-t002:** Basic properties of the SAP.

Properties	Test Results	Dry SAP	SAP Hydrogel
Dry appearance (@25 °C)	White powder	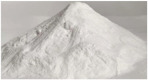	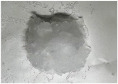
pH value (@25 °C)	6.5–7.0		
Density (g/cm^3^)	0.675		
Particle size (mesh No.)	<200		
Water absorption rate (mL/g)	>100		

**Table 3 materials-16-01082-t003:** Synthetic gradation of the asphalt mixture used in the study.

Gradation Composition	Mass Percentage (%) through the Following Sieve (mm)									
	16	13.2	9.5	4.75	2.36	1.18	0.6	0.3	0.15	0.075
Synthetic gradation	100	94.7	75.6	52.4	35.2	27.6	21.3	11.9	7.9	5.0
upper limit	100	100	85	68	50	38	28	20	15	8
lower limit	100	90	68	38	24	15	10	7	5	4

**Table 4 materials-16-01082-t004:** The density and porosity of the asphalt mixtures.

Type of Asphalt Mixture	Density (g/cm)3	Porosity (%)
SAP-0	2.413	4.34
SAP-10%	2.417	4.31
SAP-20%	2.416	4.30
SAP-30%	2.420	4.25
SAP-40%	2.421	4.25

**Table 5 materials-16-01082-t005:** Fitted equations between storage modulus and frequency.

SAP Content	Fitted Equation	R^2^
SAP-0	logG’(w) = 0.9004logw + 5.7263	R^2^ = 0.9987
SAP-10%	logG’(w) = 0.9124logw + 5.6928	R^2^ = 0.9987
SAP-20%	logG’(w) = 0.9026logw + 5.6871	R^2^ = 0.9986
SAP-30%	logG’(w) = 0.8919logw + 5.6651	R^2^ = 0.9984
SAP-40%	logG’(w) = 0.9015logw + 5.6633	R^2^ = 0.9985

**Table 6 materials-16-01082-t006:** The results from permeability test.

SAP Content (%)	h (mm)	v (mm/s)	v’	K (mm/s)
0	*h* = 0.001164t2 − 1.147t + 777 (R² = 0.9981)	*v* = −0.03231i2 + 0.8318i − 4.369 (R² = 0.9972)	*v’* = − 0.06462i + 0.8318	0.06305
10	*h* = 0.0005297t2 − 0.7714t + 778.4 (R² = 0.9977)	*v* = −0.02398i2 + 0.6077i − 3.197 (R² = 0.9968)	*v’* = − 0.04796i + 0.6077	0.04109
20	*h* = 0.0004043t2 − 0.6717t + 777.8 (R² = 0.9978)	*v* = −0.02330 i2 + 0.5599i − 2.948 (R² = 0.9956)	*v’* = − 0.04660i + 0.5599	0.03504
30	*h* = 0.0003366t2 − 0.6168t + 780.8 (R² = 0.9982)	*v* = −0.02019i2 + 0.5073i − 2.67 (R² = 0.9943)	*v’* = − 0.04038i + 0.5073	0.03164
40	*h* = 0.0002857t2 − 0.5682t + 690.3 (R² = 0.9984)	*v* = −0.01874i2 + 0.4695i − 2.47 (R² = 0.9933)	*v’* = − 0.03748i + 0.4695	0.02855

## Data Availability

All data, models, and codes generated or used in this study are included in the submitted manuscript.

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
