# Peer review of "Investigation on Performances and Functions of Asphalt Mixtures Modified with Super Absorbent Polymer (SAP)"

_materials, 2023, doi:10.3390/ma16031082_

Round 1
Reviewer 1 Report
Please have a look on the attached file

Reviewer 2 Report
Sun et al. have presented the manuscript titled: Study on the asphalt materials incorporated with super absorbent polymer (SAP). Overall presentation of the article is good, and authors have provided the detailed study. I just have few suggestions for the authors about this article.
1. Abstract is so general and weak, authors should make it precise which should only highlight their achievement (results, values), and one or two sentences of the research applications.
2. Authors are suggested to add the EDS spectra or 2D colored elemental mapping along with Figure 8.
3. Authors should also highlight the density and porosity of the samples as well.
Reviewer 3 Report
This paper has relatively complete research about SAP applying for asphalt materials. It also got some interesting results. Here are some comments to help improve reach publish requirements.
1, All the rheological figures are not clear enough. Using the software origin to draw, especially the figure in the figure, it is very easy to adjust.
2, the SAP of water absorption ratio reach 105ml/g, such a large amount of water absorption is just to block the cracks, so there is no need to replace such a large amount of mineral powder to reach 10%-40%.
3, After absorbing water, whether there will be an effect on crack expansion? so the low-temperature crack resistance of concrete and multiple freeze-thaw cycles will be more important. The author does not have more research and explanation.
Round 2
Reviewer 1 Report
Thanks for considering my comments positively. One point still left. The font and size of figures is not of that level. It will affect quality of your work.